# Higher Plasma Creatinine Is Associated with an Increased Risk of Death in Patients with Non-Metastatic Rectal but Not Colon Cancer: Results from an International Cohort Consortium

**DOI:** 10.3390/cancers15133391

**Published:** 2023-06-28

**Authors:** Jennifer Ose, Biljana Gigic, Stefanie Brezina, Tengda Lin, Anita R. Peoples, Pauline P. Schobert, Andreas Baierl, Eline van Roekel, Nivonirina Robinot, Audrey Gicquiau, David Achaintre, Augustin Scalbert, Fränzel J. B. van Duijnhoven, Andreana N. Holowatyj, Tanja Gumpenberger, Petra Schrotz-King, Alexis B. Ulrich, Arve Ulvik, Per-Magne Ueland, Matty P. Weijenberg, Nina Habermann, Pekka Keski-Rahkonen, Andrea Gsur, Dieuwertje E. Kok, Cornelia M. Ulrich

**Affiliations:** 1Huntsman Cancer Institute, Salt Lake City, UT 84112, USA; 2Department of Population Health Sciences, University of Utah, Salt Lake City, UT 84112, USA; 3Department of General, Visceral and Transplantation Surgery, University of Heidelberg, 69117 Heidelberg, Germany; biljana.gigic@med.uni-heidelberg.de (B.G.);; 4Institute of Cancer Research, Department of Medicine I, Medical University of Vienna, 23, 1090 Vienna, Austria; stefanie.brezina@meduniwien.ac.at (S.B.);; 5School of Medicine, Ludwig-Maximilians University, 80539 Munich, Germany; 6School of Medicine, Technical University of Munich, 80333 Munich, Germany; 7Department of Statistics and Operations Research, University of Vienna, 1, 1010 Wien, Austria; 8Department of Epidemiology, GROW-School of Oncology and Developmental Biology, Maastricht University, 30, 6229 Maastricht, The Netherlands; 9Nutrition and Metabolism Branch, International Agency for Research on Cancer, WHO, 69366 Lyon, France; 10Division of Human Nutrition and Health, Wageningen University & Research, 6708 Wageningen, The Netherlands; 11Department of Medicine, Vanderbilt University Medical Center, Nashville, TN 37232, USA; 12Vanderbilt-Ingram Cancer Center, Nashville, TN 37232, USA; 13Division of Preventive Oncology, National Center for Tumor Diseases (NCT) and German Cancer Research Center (DKFZ), 69120 Heidelberg, Germany; 14Klinik für Allgemein-, Viszeral-, Thorax- und Gefäßchirurgie, Städtische Kliniken Neuss, 84, 41464 Neuss, Germany; 15BEVITAL, 87, 5021 Bergen, Norway; 16Genome Biology, European Molecular Biology Laboratory (EMBL), 69117 Heidelberg, Germany

**Keywords:** all-cause mortality, creatinine, metabolites, colon cancer, rectal cancer, all-cause mortality, starch and sucrose pathway

## Abstract

**Simple Summary:**

Colorectal cancer is increasingly recognized as two separate diseases: colon cancer and rectal cancer, each with its own causes and outcomes. We included 674 patients with colorectal cancer, with pre-surgery collected blood samples, and patient follow up for an average of 4.4 years. Ninety-three patients (14%) died from various causes including 60 patients with colon cancer and 33 patients with rectal cancer. Higher levels of plasma creatinine increased the risk of death by 39% in patients with rectal but not colon cancer. We further identified a biological pathway (e.g., roadmaps inside our bodies that guide various processes) related to starch and sucrose metabolism, which was linked to worse clinical outcomes in colon cancer but not rectal cancer. There is some evidence, that resistant starch, which resists digestion in the small intestine, may offer protection against colon cancer. Understanding the distinct causes and outcomes of colon and rectal cancer is crucial for tailoring effective treatments. In conclusion, personalized treatment strategies should consider colon cancer and rectal cancer separately and are essential for improving patient outcomes in the future.

**Abstract:**

Colorectal cancer (CRC) is increasingly recognized as a heterogeneous disease. No studies have prospectively examined associations of blood metabolite concentrations with all-cause mortality in patients with colon and rectal cancer separately. Targeted metabolomics (Biocrates Absolute*IDQ* p180) and pathway analyses (MetaboAnalyst 4.0) were performed on pre-surgery collected plasma from 674 patients with non-metastasized (stage I–III) colon (*n* = 394) or rectal cancer (*n* = 283). Metabolomics data and covariate information were received from the international cohort consortium MetaboCCC. Cox proportional hazards models were computed to investigate associations of 148 metabolite levels with all-cause mortality adjusted for age, sex, tumor stage, tumor site (whenever applicable), and cohort; the false discovery rate (FDR) was used to account for multiple testing. A total of 93 patients (14%) were deceased after an average follow-up time of 4.4 years (60 patients with colon cancer and 33 patients with rectal cancer). After FDR adjustment, higher plasma creatinine was associated with a 39% increase in all-cause mortality in patients with rectal cancer. HR: 1.39, 95% CI 1.23–1.72, p_FDR_ = 0.03; but not colon cancer: p_FDR_ = 0.96. Creatinine is a breakdown product of creatine phosphate in muscle and may reflect changes in skeletal muscle mass. The starch and sucrose metabolisms were associated with increased all-cause mortality in colon cancer but not in rectal cancer. Genes in the starch and sucrose metabolism pathways were previously linked to worse clinical outcomes in CRC. In summary, our findings support the hypothesis that colon and rectal cancer have different etiological and clinical outcomes that need to be considered for targeted treatments.

## 1. Introduction

Colorectal cancer (CRC) is the third most common cancer in men and women in the U.S., with an estimated incidence of 151,030 patients [1]. An estimated 52,580 patients will succumb to this disease in 2022 [1]. Most observational studies in this area focus on prognostic outcomes combining colon and rectal cancer patients, often due to small sample sizes for patients with rectal cancer. However, there is mounting evidence that colon cancer and rectal cancer are distinct tumor entities with different etiological and prognostic pathways [2]. We have previously shown that patients with high *Fusobacterium nucleatum* abundance had a 5-fold increased risk of being diagnosed with rectal cancer compared with right-sided colon cancer [3]. There are further differences in the role of inflammation in colon and rectal cancer. In our study on systematic inflammation, we showed significant heterogeneity between inflammatory biomarkers and clinical outcomes (e.g., recurrence) in stratified analysis by tumor site [4]. For example, VEGF-D was associated with a 3-fold increase in the risk of death for rectal cancer (HR_log2_: 3.26; 95% CI, 1.58–6.70) compared with no association for colon cancer (HR_log2_: 0.78; 95% CI, 0.35–1.73; p_heterogeneity_ = 0.01) [4]. Besides these biological markers, there are differences in the associations of modifiable lifestyle factors, such as physical activity, obesity, or diet, between patients with rectal cancer compared with colon cancer [5,6]. The results from these prior studies support the hypothesis that tumors in different anatomical regions (colon versus rectum) have distinct etiologies and differences with regard to prognostic outcomes.

Metabolomics is a sophisticated approach to semi-quantifying many metabolites in different biospecimens, having the ability to represent the patho-physiological state of an individual [7,8,9,10,11,12]. For example, previous studies examined blood metabolic profiles in association with CRC [13,14,15,16,17,18]. Weaknesses in these studies include small sample sizes, being retrospective in nature, and not distinguishing between colon and rectal cancer. Here we present the largest prospective study to date investigating associations of metabolites and biological pathways with all-cause mortality in prospectively followed patients with colon and rectal cancer, separately.

## 2. Materials and Methods

### 2.1. Study Population

Data from four international cohort studies embedded in the metabolomic profiles throughout the Continuum of Colorectal Cancer (MetaboCCC) consortium were used for the present study [14]. MetaboCCC is a large European consortium investigating metabolic profiles across the continuum of colorectal cancer carcinogenesis [19]. The cohorts included in this study consist of: the COLON study, conducted in the Netherlands [20]; the EnCoRe study, also conducted in the Netherlands and registered under the Netherlands Trial Register with the identifier 7099 [21]; the Heidelberg site of the ColoCare Study, registered under ClinicalTrials.gov with the identifier NCT02328677 [22]; and the Colorectal Cancer Study of Austria (CORSA) [23]. These cohorts were selected for their relevance to the research and their contribution to the comprehensive analysis conducted in this study.

The research adhered to the Declaration of Helsinki guidelines and received approval from the Institutional Review Board or Ethics Committee of all participating cohorts. The ColoCare study received approval from the ethics committee at the University of Heidelberg (310/2001), while the CORSA study was approved by the ethical review committees of the Medical University of Vienna (1160/2016), the “Ethikkommission der Stadt Wien” (06-150-VK), and the institutional review board “Ethikkommission Burgenland”. Additionally, the International Agency for Research on Cancer’s ethics committee also approved the study. The COLON study received ethical approval from the Committee on Research involving Human Subjects, region Arnhem-Nijmegen (Commissie Mensgebonden Onderzoek—CMO, region Arnhem Nijmegen). Finally, the EnCoRe study was approved by the Medical Ethics Committee at the University Hospital Maastricht and Maastricht University in the Netherlands.

In total, 674 CRC patients were included in the current study, including 192, 206, 227, and 55 of the COLON, EnCoRe, ColoCare, and CORSA cohorts, respectively. We did not include three patients for stratified analysis by tumor site due to missing information on tumor sites; however, these were included for overall analysis (colon and rectal cancer combined; see Appendix A).

The cohorts included in this study are briefly introduced below. For more detailed information, please refer to the previously published manuscript [14] and the study design manuscripts specific to each cohort. COLON study: a prospective cohort study conducted in the Netherlands, with recruitment commencing in 2010 [20]; EnCoRe study: an ongoing prospective cohort study [21]; ColoCare Study: an ongoing international multi-center prospective study that commenced in 2007 [22]; CORSA: an ongoing cohort study that recruits CRC patients in collaboration with the Burgenland Prevention Trial of Colorectal Disease with Immunological Testing (B-PREDICT) project since 2003 [23]. Additional recruitment takes place at four hospitals in Vienna. The inclusion and exclusion criteria for each cohort have been previously published [20,21,22,23]. All patients included in this study had histologically confirmed non-metastatic CRC and provided pre-surgery plasma samples for targeted metabolomics analysis. Patients diagnosed with stage I/II/III CRC were eligible for inclusion. For the purpose of this study, tumor location was classified as either colon (cecum, appendix, ascending colon, hepatic flexure, transverse colon, splenic flexure, descending colon, and sigmoid colon) or rectal (rectosigmoid junction and rectum) cancer.

### 2.2. Data Collection

EDTA plasma samples were obtained from participants in all cohorts at the time of recruitment, which typically occurred shortly after the diagnosis of colorectal cancer (CRC) and predominantly before any surgical, chemotherapy, or radiotherapy treatment. The plasma samples were generally collected and processed within four hours and subsequently stored at the respective study sites at a temperature of −80 °C. Clinical data, including TNM-stage, tumor site, and treatment details such as surgery, neo-adjuvant and adjuvant chemotherapy, and/or radiotherapy, were extracted from medical records or relevant registries for all cohorts. The following demographic and lifestyle characteristics were considered as part of this study: age at diagnosis, sex, body mass index (BMI), smoking status (never, former, current), and alcohol intake in the past year (yes/no). All clinical, demographic, and lifestyle data were harmonized across the cohorts and are included in the data warehouse of the MetaboCCC Consortium.

### 2.3. Sample Analysis

In this study, non-fasting blood samples were chosen from all cohorts based on specific criteria: individuals diagnosed with non-metastasized colorectal cancer (CRC) at stages I–III, and baseline EDTA plasma available (collected after diagnosis but before treatment). All samples were transported on dry ice to the International Agency for Research on Cancer (IARC) in Lyon, France.

Targeted metabolomics analyses were conducted using the AbsoluteIDQ p180 kit by BIOCRATES Life Sciences AG, located in Innsbruck, Austria. The kit allowed for the semi-quantification of up to 188 metabolites, including 21 amino acids, 21 biogenic amines, 90 glycerophospholipids, 15 sphingomyelins, 40 acylcarnitines, and the sum of hexose sugars [24,25]. The analytical process, which involved ultra-high-performance liquid chromatography (UHPLC) coupled with tandem mass spectrometry (MS/MS), followed the manufacturer’s recommendations. Amino acids and biogenic amines were quantified using UHPLC-MS/MS, while lipids, sugars, and acylcarnitines were analyzed using flow injection (FIA)-MS/MS. Detailed information regarding the analytical procedure can be found in previous publications [14,26].

For the selected 148 metabolites, imputation techniques were employed to replace missing values that accounted for less than 20% across all cohorts. The following procedures were implemented: for values below the limit of detection (LOD), imputation was performed using half of the batch-specific LOD, while values below the lower limit of quantification (LOQ) were imputed with the LOQ value. Values above the upper LOQ were set at the upper limit. These imputation procedures were chosen to align with the approaches utilized in previous studies that employed data from the same kit.

These are the metabolites in the respective classes: 11 acylcarnitines, 19 amino acids, 7 biogenic amines, hexose, 10 lysophosphatidylcholines, 34 phosphatidylcholines (acyl-alkyl), 35 glcyerophospholipids, 17 carnitine, and 14 sphingolipids. Metabolites that had missing values in more than 50% of either the deceased patients (cases) or the alive patients (controls) were excluded from the analysis. Any remaining missing values, up to a maximum of 50%, were not imputed following the guidelines outlined by Di Guida et al. [27]. An overview of all measured metabolites can be found in Appendix A. All metabolites were linked to their respective Human Metabolome Database (HMDB) identifiers, which are required for pathway analysis using the MetaboAnalyst software 4.0 [28]. The utilized pathway analysis uses available pathway databases and known gene expression data to identify the pathways that are significantly impacting a given condition, in this case, all-cause mortality in patients diagnosed with CRC.

### 2.4. Study Endpoint

The primary study endpoint was all-cause mortality in patients with newly diagnosed non-metastatic CRC. Diagnosis was confirmed by pathology. Follow-up time was calculated starting from the date of blood collection until the date of either the last follow-up or the date of death, whichever occurred first. For the COLON study, all-cause mortality data Information regarding vital status, such as whether an individual is deceased or alive and the date of death, is collected using different sources depending on the specific study [20]. In the ColoCare Study, vital status is obtained through various means, including medical records, regular mailings for follow-up, periodic requests for external medical records, and state or national cancer and death registries [22]. In the Austrian cohort study (CORSA), clinical data, including vital status, is abstracted from medical records and entered into a structured database following standardized documentation guidelines, in compliance with the General Data Protection Regulation [23]. In the ENCORE study, mortality data, including information on whether an individual is deceased or alive and the date of death, were retrieved from the Municipal Personal Records Database [21]. 

### 2.5. Statistical Analysis

Prior to statistical analysis, the intensities of metabolites were logarithmically transformed to a base of 2 in order to address heteroscedasticity. Demographic and clinical characteristics were presented as medians with corresponding ranges or as numbers accompanied by percentages for categorical variables. Body mass index (BMI) was calculated by dividing weight in kilograms by the square of height in meters (kg/m²). BMI status was categorized according to the World Health Organization (WHO) guidelines as follows: underweight (<18.5 kg/m²), normal weight (≥18.5–24.9 kg/m²), overweight (25.0–29.9 kg/m²), and obese (≥30.0 kg/m²). Smoking status was classified into three categories: current, former, and never. Further, alcohol intake in the past year before diagnosis (yes and no) has been documented. For the present study, we are focusing on differences in associations of metabolites with all-cause mortality in colon and rectal cancer, separately. Additionally, an overall analysis (combining colon and rectal cancer) was conducted and is available in the Appendix A.

Cox proportional hazards regression models were used to assess the relationship between metabolites and all-cause mortality. The HR represents the change in risk for all-cause mortality caused by an increase of one standard deviation (SD) in metabolite intensity, allowing comparison of effect sizes between different metabolites. Heterogeneity in associations between biomarkers and clinical outcomes stratified by aforementioned factors (e.g., tumor site and study site) was assessed using likelihood-ratio tests for the comparison of the model fit for logistic regression models with and without corresponding interaction terms. In an additional analysis, we did not identify significant differences in all-cause mortality comparing models adjusted for BMI to models not adjusted for BMI. Equally, adjustment for adjuvant treatment did not change the overall model. We did not adjust the analysis by receipt of neo-adjuvant treatment since this is a proxy for adjustment by tumor site, as predominantly rectal cancer patients receive neo-adjuvant treatment. We have compared results across cohorts and observed very similar results (Appendix A), but finally decided to adjust all analyses by study site. The final model was adjusted for age, sex, disease stage, tumor site, and cohort; stratified analyses by tumor site were not adjusted for tumor site. We provide information on the *p*-value for each of the analyses. Given the associations of creatinine with all-cause mortality, we performed an additional analysis to adjust all models with creatinine as a confounder. This additional analysis reported similar results compared with those not adjusted for creatinine. All analyses were computed in SAS 9.4 (Cary, NC, USA). 

*A priori*, an FDR q-value < 0.05 was considered statistically significant. Analysis comparing colon and rectal cancer separately was conducted and is presented in forest plots by using the package ‘metafor’ in R, version 3.3.6.

The relevant pathways were identified using the pathway analysis module of MetaboAnalyst 4.0 [29]. This module combines results from pathway enrichment analysis with pathway topology analysis to help identify the most relevant pathways involved in colon and rectal cancer [29]. We first included all metabolites that were associated with all-cause mortality for colon and rectal cancer combined (Appendix A). Subsequently, we performed additional analyses: (1) limited to metabolites associated with increased risk of all-cause mortality; and (2) limited to metabolites associated with reduced risk of all-cause mortality. These additional analyses provide evidence on which pathways are relevant for an increased risk of death and which ones are related to a reduced risk of death, enabling a better understanding of the role of the respective pathways in clinical outcomes.

Creatinine was the only metabolite associated with all-cause mortality after FDR adjustment. Thus, we decided to test for linearity in the association of creatinine with all-cause mortality, as a previous study observed a non-linear relationship between creatinine and all-cause mortality [30]. In order to investigate the relationship between creatine and all-cause mortality, we plotted the creatine values against martingale residuals obtained from multiple Cox regression (adjusting for study cohort, age, sex, and tumor stage). There is no parent pattern regarding the creatine values and martingale residuals; thus, we conclude that the linearity assumption is held. Thus, no additional nonlinear regression analysis was performed.

## 3. Results

### 3.1. Study Population Characteristics

For the present study, we included 674 colon and rectal cancer patients from the international cohort consortium MetaboCCC. Table 1 shows the patient population stratified by tumor site. Patients had a median (range) age at diagnosis of 66 (60–73) years and were predominantly male (65%). A total of 394 patients were diagnosed with colon cancer (58%), and 283 patients were diagnosed with rectal cancer (42%). All patients had undergone surgery. Twenty-seven percent of patients received neo-adjuvant therapy, and 31% of patients received adjuvant therapy. After a follow-up time of 4.8 years (3.16–6.03 years), 93 patients (14%) were deceased. The descriptive statistics across prospective cohorts are described separately in Appendix A.

### 3.2. Associations of Metabolites with All-Cause Mortality

#### 3.2.1. Associations of Metabolites with All-Cause Mortality in Patients with Colon Cancer

In analyses limited to patients with colon cancer, we observed associations between seven metabolites and all-cause mortality (Figure 1). We observed inverse associations for two metabolites: histidine: hazard ratio (HR): 0.72, 95% Confidence Interval (CI): 0.54, 0.96, *p* = 0.02, p_FDR_ = 0.61. An increased risk of death was observed for five metabolites. These include proline: HR: 1.34: 95% CI: 1.07–1.68, *p* < 0.001, p_FDR_ = 0.49, and hexose: HR: 1.39, 95% CI: 1.12, 1.74, *p* < 0.001, and p_FDR_ = 0.26, although not significant after FDR adjustment.

#### 3.2.2. Associations of Metabolites with All-Cause Mortality in Patients with Rectal Cancer

In analyses limited to rectal cancer, we observed associations of 39 metabolites with all-cause mortality (Figure 2). We observed inverse associations for 34 metabolites, including glycerophospholipids and sphingomyelins: PC:ae:C30:2: HR: 0.41, 95% CI: 0.24–0.71, *p* < 0.01, and p_FDR_ = 0.07. For five metabolites, we observed an increased risk of death, although it was not significant after FDR adjustment. For creatinine, we observed a 39% increase in the risk of all-cause mortality after FDR adjustment: HR: 1.39, 95% CI: 1.23–1.72, p_FDR_ = 0.03. The associations of metabolites with all-cause mortality in the entire study population are provided in Appendix A.

#### 3.2.3. Heterogeneity in the Associations of Metabolites with All-Cause Mortality in Patients with Rectal Cancer Compared with Colon Cancer

We performed heterogeneity tests to investigate if associations of metabolites with all-cause mortality are significantly different by anatomical site. Out of 148 metabolites, heterogeneity was evident for 13 metabolites, including 11 glycerophospholipids (range for p_heterogeneity_: <0.007–0.03) and two lysophospoline metabolites (p_heterogeneity_: 0.03, 0.04, respectively). We did not observe heterogeneity for associations of creatine with all-cause mortality (p_heterogeneity_ = 0.11; Table 2).

### 3.3. Pathway Analysis

#### 3.3.1. Pathway Analysis in Patients with Colon Cancer

We performed pathway analysis for all patients diagnosed with colon cancer (Table 3). A total of 10 pathways were identified, including arginine and proline, sphingolipids, and beta-alanine. For metabolites inversely associated with all-cause mortality (Table 3), five pathways were identified, such as the sphingolipid pathway and the beta-alanine pathway. We observed five significant pathways when limiting the dataset to metabolites that were associated with an increased risk of death, such as arginine and proline, pentose, or the starch and sucrose pathway (Table 3).

#### 3.3.2. Pathway Analysis in Rectal Cancer

For rectal cancer, the most relevant pathway was glycerophospholipid metabolism, which was associated with reduced all-cause mortality [31] (Table 4).

## 4. Discussion

This study is the largest prospective study to date investigating associations between plasma metabolites and all-cause mortality in patients diagnosed with colon and rectal cancer separately.

Creatinine was associated with all-cause mortality in patients with CRC after FDR adjustment. In stratified analysis by anatomical location, higher levels of creatinine were associated with a 39% increase in the risk of all-cause mortality for patients diagnosed with rectal cancer but not colon cancer.

Different biological pathways showed up for patients diagnosed with colon cancer compared with patients diagnosed with rectal cancer. For example, colon cancer was linked to the arginine and proline metabolism pathways and glycolysis/gluconeogenesis, while these were not related to all-cause mortality in patients with rectal cancer. Five pathways were inversely associated with the risk of all-cause mortality in patients with colon cancer, e.g., glycerophospholipid metabolism or linoleic acid metabolism. The beta-alanine pathway was inversely associated with all-cause mortality in colon cancer but not in rectal cancer. The starch and sucrose metabolisms were associated with all-cause mortality overall and in patients diagnosed with colon cancer but not with rectal cancer.

Creatinine plays an essential role in tissues with high metabolic demand, such as muscle tissue. Creatinine production relies on the size of the creatine pool, which is determined by total muscle mass and dietary intake of meat [32]. Over 90% of total creatine is stored in skeletal muscles, where it is converted into creatinine and released into the circulation [32]. Interestingly, serum creatinine has been shown to be a potential prognostic biomarker in numerous cancer types, including CRC. In a large retrospective study including *n* = 1733 CRC patients, a non-linear association between serum creatinine and overall all-cause mortality in CRC was observed [30]. For example, low or high serum creatinine concentrations were associated with significantly increased all-cause mortality compared with patients with normal concentrations [30]. Notably, one of the major limitations of this study was that laboratory data were obtained from common laboratory tests in the clinic and were limited in scope. In the present study, the association of creatinine was measured with a state-of-the-art metabolomics assay, which is an advantage over prior investigations.

One prior study showed that systemic creatinine concentrations are linked to all-cause mortality in myofibroblast and fibroplastic sarcomas [33]. In this study of *n* = 132 cancer patients, elevated serum creatinine levels were significantly associated with a 3-fold increased risk of impaired sarcoma-specific survival, even after adjustment for tumor stage at diagnosis (subdistribution HR per 1 mg/dL increase: 3.27; 95% CI: 1.87–5.73; *p* < 0.0001). There have been further studies in renal carcinoma and epithelial ovarian cancer confirming this observation [34,35]. The study on renal cancer included *n* = 230 patients, and pre-treatment serum creatinine was associated with worse 5-year overall survival in this retrospective cohort [34]. In another retrospective study including *n* = 498 patients diagnosed with epithelial ovarian cancer, serum levels of creatinine were statistically significantly associated with mortality, independent of prognostic parameters for overall survival [35]. One underlying biological mechanism of these observations may be the release of creatinine from the damaged muscle, which may be an indicator of muscle lysis, which may lead to cachexia in cancer patients. The higher the muscle lysis, the higher the creatinine concentrations in the circulation [36] which may explain the present results with all-cause mortality.

Notably, creatinine was associated with rectal but not colon cancer. One of the major differences between colon and rectal cancer patients is the receipt of neo-adjuvant treatment (e.g., radiotherapy and chemotherapy) for patients with rectal cancer to shrink the tumor and enable surgery [37]. Commonly prescribed neo-adjuvant chemotherapy agents include 5-fluorouracil (5-FU) and oxaliplatin. Oxaliplatin acts via the formation of DNA-platinum adducts, which deprive tumor cells of the necessary building blocks for cell replication. There is in vivo evidence that treatment with oxaliplatin leads to increased muscle lysis in mice [38]. Thus, the observed associations between creatine and rectal cancer may be linked to the treatment with oxaliplatin, which leads to muscle lysis and subsequently increased creatinine concentrations [38]. As patients with colon cancer usually do not receive this type of pre-surgery treatment, there may be no impact on the creatinine levels in patients with colon cancer, as demonstrated in the present body of work.

For the discussion of the identified pathways related to colon cancer, we will describe exemplary pathways: the starch and sucrose metabolism pathway (increased risk of all-cause mortality) and the beta-alanine pathway (decreased risk of all-cause mortality) [39]. The starch and sucrose metabolism pathway has a crucial role in tumor progression. In previous studies, this pathway was associated with the progression of colon cancer [40]. Notably, resistant starch (a carbohydrate that resists digestion in the small intestine and ferments in the large intestine) in the diet may prevent carcinogenesis of colon epithelial cells [40]. The described mechanism involves enhancing cell death (apoptosis) by inducing stress in the endoplasmic reticulum, leading to a pathway of mitochondrial apoptosis [40]. Beta-alanine, an amino acid that can be produced within the body, is converted into carnosine, which acts as a buffer within cells. The accumulation of carnosine (β-alanyl-L-histidine) has been demonstrated to decrease the growth of tumor cells in laboratory cultures and inhibit tumor development in living organisms [41,42,43]. It is proposed that carnosine accumulation restricts glycolysis, thereby limiting energy production [43]. Moreover, carnosine has been observed to reduce the production of cellular adenosine triphosphate (ATP) in glioma cells [44]. Although there are no data yet on colorectal cancer, it is suggested that the benefits of carnosine in suppressing tumors are, in part, due to the inhibition of glycolysis [41,42,43]. In a previous study in epithelial breast cells, β-alanine reduced cancerous metabolism and extracellular acidification, which is known to suppress tumor aggressiveness [45]. Data from an in vitro study demonstrated that carnosine has the potential to suppress human CRC cell proliferation [43]. The authors discussed that reducing the β-catenin/Tcf-4 signaling can induce necroptosis and autophagy, thereby inhibiting angiogenesis [43]. Some metabolites, such as carnosine, have direct links with dietary consumption, e.g., carnosine is present in fish. If the present findings are replicated by future studies, the potential role of specific dietary recommendations could be further explored regarding fish consumption, e.g., yellowfin tuna and eel.

For rectal cancer, glycerophospholipid metabolism was associated with reduced all-cause mortality. The role of glycerophospholipids in carcinogenesis is only partially understood [46]. One explanation could be that glycerophospholipids have anti-inflammatory properties and thereby protect from oxidative stress, which leads to inhibition of cell proliferation and induces apoptosis [46]. Our results suggested that concentrations of glycerophospholipids (phosphatidylcholines and lysophosphatidylcholines) are linked to an inverse association of glycerophospholipids with all-cause mortality in patients with rectal cancer, which is in line with the evidence provided here.

This study has strengths and limitations. An important strength of the current study is the large sample size that enables us to investigate associations between circulating plasma metabolites and all-cause mortality in colon and rectal cancer separately. Creatinine is an established marker of renal function; thus, the observed associations with all-cause mortality may be reflective of the prevalence of kidney failure in our study population. Unfortunately, the participating cohorts did not collect information on renal failure. The prospective study design can provide high-quality data on the primary exposure and a large set of confounding variables, which reduces error due to unmeasured confounding. We performed stratified analyses by cohort to ensure results were comparable, and we observed similar results. For the present study, we do not have access to information on whether the cause of death was cancer related. Fasting status has been previously shown to impact the measurement of certain metabolites such as acylcarnitines and phosphatidylcholines [25,47,48], but this is not impacting the metabolites that were identified in this study. Diet, specifically red meat, is known to impact creatinine metabolism. It is expected that meat consumption is similar across colon and rectal cancer patients. In this case, that would be defined as non-differential misclassification, which causes an underestimate of the true association.

## 5. Conclusions

In summary, the study suggests that creatinine can serve as a valuable biomarker for predicting all-cause mortality in rectal cancer patients. The findings also highlight the need to consider differences in biological pathways and prognostic factors between colon and rectal cancers. In the future, researchers should focus on identifying additional biomarkers and pathways associated with the prognosis of these cancers to develop personalized treatment strategies tailored to each patient’s unique needs. A better understanding of the molecular mechanisms driving colon and rectal cancer progression and mortality can lead to improved patient outcomes and a reduced burden of these devastating diseases in colorectal cancer patients.

## Figures and Tables

**Figure 1 cancers-15-03391-f001:**
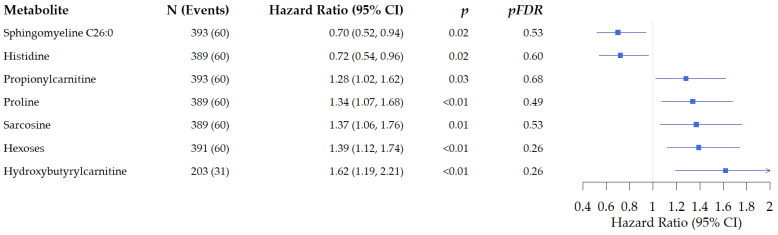
Associations of metabolites with all-cause mortality in patients diagnosed with colon cancer. Presented are the metabolites that were significant before FDR adjustment.

**Figure 2 cancers-15-03391-f002:**
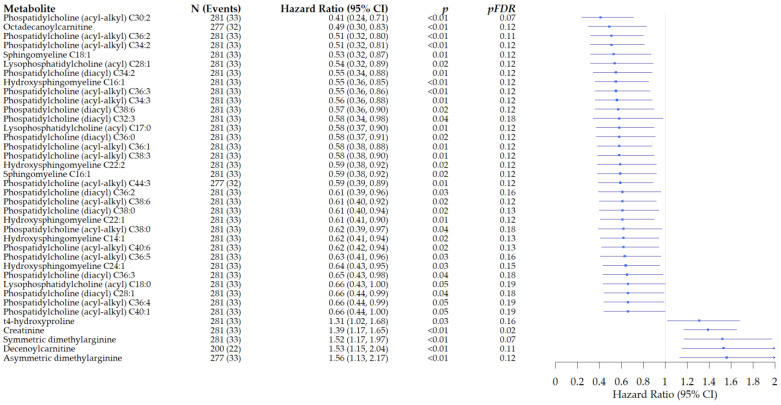
Associations of metabolites with all-cause mortality in patients diagnosed with rectal cancer. Presented are the metabolites that were significant after FDR adjustment.

**Table 1 cancers-15-03391-t001:** Baseline characteristics of the total study population of MetaboCCC stratified by tumor site.

	Patients with Colon Cancer	And with Rectal Cancer	*p*-Value *
	Colon cancer	Rectal cancer	
**Number of participants, n (%)**	393 (58)	281 (42)	
**Age at diagnosis**, years (median, range)	68 (62–75)	63 (56–70)	**<0.01**
**Sex**, n (%)			
Male	243 (62)	194 (69)	0.05
Female	150 (38)	87 (31)	
**Vital status**, n (%)			
Alive	333 (85)	248 (88)	0.19
Deceased	60 (15)	33 (12)	
**Follow-up time**, years (median, range)			
Alive	4.82 (3.16–6.03)	4.01 (2.58–5.61)	**<0.001**
Deceased	2.79 (1.29–4.45)	2.84 (0.92–3.51)	
**Stage of disease**, n (%)			
I	118 (30)	55 (20)	**<0.01**
II	152 (39)	62 (22)	
III	123 (31)	164 (58)	
**Neo-adjuvant treatment**, n (%)			
Yes	4 (1)	176 (63)	**<0.01**
No	389 (99)	105 (37)	
**Surgery**, n (%) **			
Yes	392 (99.7)	271 (96)	**<0.01**
No	1 (0.3)	10 (4)	
**Adjuvant treatment**, n (%)			
Yes	116 (30)	90 (33)	0.45
No	268 (70)	183 (67)	
**Body mass index**			
**Continuous**, kg/m^2^ (median, range)	26.80 (24.20–30.00)	26.30 (24.00–29.10)	**0.048**
Underweight, <18.5, n (%)	3 (1)	4 (1)	0.23
Normal weight, 18.5–24.9, n (%)	122 (31)	98 (35)	
Overweight, 25–29.9, n (%)	168 (43)	125 (45)	
Obese, ≥30, n (%)	100 (25)	54 (19)	
**Height**, m (median, range)	1.72 (1.65–1.78)	1.73 (1.66–1.79)	0.06
**Weight**, kg (median, range)	80.00 (69.5–90.00)	79.60 (70.00–88.00)	0.07
**Smoking**, n (%)			
Current	43 (11)	58 (21)	**<0.01**
Former	194 (51)	144 (53)	
Never	144 (38)	71 (26)	
**Alcohol intake, n (%)**			
**Yes**	325 (85%)	247 (89%)	0.14
**No**	58 (15%)	31 (11%)	

* All *p*-values that are bolded are statistically significant. ** Analyses were not adjusted for surgery as all participants received surgery.

**Table 2 cancers-15-03391-t002:** Significant heterogeneity in associations of metabolites with colon and rectal cancer *.

	HR 95% CI	HR 95% CI	p_Interaction
Metabolite Name	Colon Cancer	Rectal Cancer	
PC_ae_C36_3	1.01 (0.77, 1.33)	0.55 (0.36, 0.86)	0.007
PC_aa_C36_3	1.11 (0.82, 1.49)	0.65 (0.43, 0.98)	0.008
PC_ae_C38_3	1.04 (0.79, 1.38)	0.58 (0.38, 0.90)	0.01
PC_ae_C34_2	0.94 (0.72, 1.24)	0.51 (0.32, 0.81)	0.01
PC_ae_C36_2	0.96 (0.73, 1.28)	0.51 (0.32, 0.80)	0.01
PC_aa_C38_3	1.21 (0.91, 1.62)	0.73 (0.47, 1.13)	0.01
PC_aa_C36_2	1.05 (0.77, 1.43)	0.61 (0.39, 0.96)	0.02
PC_ae_C44_3	0.80 (0.58, 1.10)	0.56 (0.36, 0.88)	0.02
PC_ae_C36_1	1.01 (0.76, 1.34)	0.58 (0.38, 0.88)	0.02
PC_ae_C30_2	0.91 (0.68, 1.22)	0.41 (0.24, 0.71)	0.03
PC_aa_C34_2	0.96 (0.71, 1.31)	0.55 (0.34, 0.88)	0.03
lysoPC_a_C18_0	0.96 (0.70, 1.30)	0.66 (0.43, 1.00)	0.03
lysoPC_a_C17_0	0.90 (0.66, 1.23)	0.58 (0.38, 0.90)	0.045

* Presented are the metabolites that were significantly different between colon and rectal cancer before FDR adjustment.

**Table 3 cancers-15-03391-t003:** Pathway analysis for metabolites associated with all-cause mortality in patients diagnosed with colon cancer.

Pathway Analysis	Total	Expected	Hits	FDR	Impact
Arginine and proline metabolism	77	0.19	2	<0.001	0.02
Sphingolipid metabolism	25	0.06	1	<0.001	0.01
Beta-alanine metabolism	28	0.07	1	<0.001	<0.01
Glycolysis or Gluconeogenesis	31	0.08	1	<0.001	<0.01
Pentose phosphate pathway	32	0.08	1	<0.001	<0.01
Nitrogen metabolism	39	0.10	1	<0.001	<0.01
Histidine metabolism	44	0.11	1	<0.001	0.14
Glycine, serine, and threonine metabolism	48	0.12	1	<0.001	0.05
Starch and sucrose metabolism	50	0.12	1	<0.001	<0.01
Aminoacyl-tRNA biosynthesis	75	0.19	1	<0.001	<0.01
**Pathways related to metabolites associated with increased risk of death**					
Arginine and proline metabolism	77	0.13	2	<0.001	0.02
Glycolysis or Gluconeogenesis	31	0.05	1	<0.001	<0.01
Pentose phosphate pathway	32	0.05	1	<0.001	<0.01
Glycine, serine, and threonine metabolism	48	0.08	1	<0.001	0.05
Starch and sucrose metabolism	50	0.08	1	<0.001	<0.01
**Pathways related to metabolites associated with decreased risk of death**					
Sphingolipid metabolism	25	0.02	1	<0.001	0.01
Beta-alanine metabolism	28	0.02	1	<0.001	<0.01
Nitrogen metabolism	39	0.03	1	<0.001	<0.01
Histidine metabolism	44	0.04	1	<0.001	0.14
Aminoacyl-tRNA biosynthesis	75	0.06	1	<0.001	<0.01

**Table 4 cancers-15-03391-t004:** Pathway analysis for metabolites associated with all-cause mortality in patients diagnosed with rectal cancer.

Pathway Analysis Overall	Total	Expected	Hits	*p*-Value	Impact
Glycerophospholipid metabolism	39	0.10	2	<0.001	0.10
Arginine und proline metabolism	77	0.19	2	<0.001	0.08
Linoleic acid metabolism	15	0.04	1	<0.001	<0.01
Sphingolipid metabolism	25	0.06	1	<0.001	0.01
Alpha-linolenic acid metabolism	29	0.07	1	<0.001	<0.01
Arachidonic acid metabolism	62	0.15	1	<0.001	<0.01
**Pathways related to metabolites associated with increased risk of death**					
Arginine und proline metabolism	77	0.10	2	<0.001	0.08
**Pathways related to metabolites associated with decreased risk of death**					
Glycerphospholipid metabolism	39	0.05	2	<0.001	0.10
Linoleic acid metabolism	15	0.02	1	<0.001	<0.01
Sphingolipid metabolism	25	0.03	1	<0.001	0.01
Alpha-linolenic acid metabolism	29	0.04	1	<0.001	<0.01
Arachidonic acid metabolism	62	0.08	1	<0.001	<0.01

## Data Availability

The data described in the manuscript, code book, and analytic code have been generated from European-based consortia. Therefore, the generated data are subject to regulations from multiple European countries, which limit our availability to share data. The consortium’s funding has ended, and no centralized staff is available to support data requests. However, the MetaboCCC PIs have agreed to answer any queries or discuss potential projects with anyone interested in future collaborative research. For further questions, please contact colocarestudy_admin@hci.utah.edu.

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
