# Peer review of "Higher Plasma Creatinine Is Associated with an Increased Risk of Death in Patients with Non-Metastatic Rectal but Not Colon Cancer: Results from an International Cohort Consortium"

_cancers, 2023, doi:10.3390/cancers15133391_

Round 1

Reviewer 1 Report

The discussion of the difference between colon cancer and colorectal cancer is good. I learned something.

However I had a major problem with the discussion on lines 72 to 75. "In our study on systemic inflammation, we showed significant heterogeneity between inflammatory biomarkers and clinical outcome ..."

My concern was the total lack of details or a citation where you published this information. Also the word "inflammation" was not to be seen in the rest of the paper.

Author Response

Reviewer 1

The discussion of the difference between colon cancer and colorectal cancer is good. I learned something.

              We thank the reviewer for this great feedback to our manuscript.

However, I had a major problem with the discussion on lines 72 to 75. "In our study on systemic inflammation, we showed significant heterogeneity between inflammatory biomarkers and clinical outcome ..." My concern was the total lack of details or a citation where you published this information. Also, the word "inflammation" was not to be seen in the rest of the paper.

Thank you very much for this feedback. We have added the respective reference on our previously published paper. The heterogeneity in the associations of inflammatory biomarkers with colon and rectal cancer was mentioned in this context, as it adds to the evidence base to perform analysis in colon and rectal cancer separately. The following text and references have been added to the introduction section of this manuscript:

In our prior study, we showed significant heterogeneity between systemic biomarkers and clinical outcomes (e.g., overall survival) in stratified analysis by tumor site[4].For example, VEGF-D was associated with a 3-fold increase in the risk of death for rectal cancer (HRlog2: 3.26; 95% CI, 1.58-6.70) compared to no association for colon cancer (HRlog2: 0.78; 95% CI, 0.35-1.73; pheterogeneity = 0.01) [4]. (Introduction)

Reviewer 2 Report

I would like to congratulate the authors on their fascinating work regarding this interesting article entitled "Higher plasma creatinine is associated with an increased risk of death in patients with non-metastatic rectal but not colon cancer: Results from an international cohort consortium ". The manuscript is well-written and the incorporated tables make the study easy to follow.

I strongly recommend acceptance for publication of the paper after minor revision.

1) I would like a brief discussion on colorectal cancer procedures and their complications. According to the literature, following colorectal cancer procedures, postoperative complications such as wound infections and sepsis are significantly more common among patients over 65 years old, with an ASA score > 2, and also with associated comorbidities such as diabetes and cardiovascular disease.

These complications have been associated with negative economic impact, increased morbidity, extended postoperative hospital stay, readmission, sepsis, and death.

I would suggest adding this important information to the discussion section and consider citing the recently published articles

https://pubmed.ncbi.nlm.nih.gov/35371356/

Author Response

Reviewer 2

I would like to congratulate the authors on their fascinating work regarding this interesting article entitled "Higher plasma creatinine is associated with an increased risk of death in patients with non-metastatic rectal but not colon cancer: Results from an international cohort consortium ". The manuscript is well-written and the incorporated tables make the study easy to follow. I strongly recommend acceptance for publication of the paper after minor revision.

              We thank the reviewer for his/her kind feedback of our manuscript. It is very much appreciated.

1) I would like a brief discussion on colorectal cancer procedures and their complications. According to the literature, following colorectal cancer procedures, postoperative complications such as wound infections and sepsis are significantly more common among patients over 65 years old, with an ASA score > 2, and also with associated comorbidities such as diabetes and cardiovascular disease. These complications have been associated with negative economic impact, increased morbidity, extended postoperative hospital stay, readmission, sepsis, and death.

I would suggest adding this important information to the discussion section and consider citing the recently published articles https://pubmed.ncbi.nlm.nih.gov/35371356/.

These are great suggestions and will help improving the discussion section of the manuscript. We have added the following text to the discussion and are citing the manuscript that was recommended from the reviewer.

We would like to point out that there are colorectal cancer treatments and related complications, which have an impact on survival. This is particularly true for medical procedures following colorectal cancer diagnosis including postoperative complications (e.g., infections and sepsis) which are also associated with comorbidities such as diabetes or cardiovascular disease. These complications have been recently associated with increased morbidity, sepsis, and death.[55] For the present body of work the respective data on sepsis, cardiovascular disease etc. are not available. However, we have excluded patients who deceased within 30 days after surgery, to account for death related to postoperative complications. (Discussion)